# Cryo-EM structure of cardiac amyloid fibrils from an immunoglobulin light chain AL amyloidosis patient

Paolo Swuec [1,2], Francesca Lavatelli[3], Masayoshi Tasaki[3,4,5], Cristina Paissoni[1], Paola Rognoni[3], Martina Maritan[1], Francesca Brambilla[6], Paolo Milani[3], Pierluigi Mauri [6], Carlo Camilloni [1], Giovanni Palladini[3], Giampaolo Merlini[3], Stefano Ricagno [1] & Martino Bolognesi [1,2]

Systemic light chain amyloidosis (AL) is a life-threatening disease caused by aggregation and deposition of monoclonal immunoglobulin light chains (LC) in target organs. Severity of heart involvement is the most important factor determining prognosis. Here, we report the 4.0 Å resolution cryo-electron microscopy map and molecular model of amyloid fibrils extracted from the heart of an AL amyloidosis patient with severe amyloid cardiomyopathy. The helical fibrils are composed of a single protofilament, showing typical 4.9 Å stacking and cross-β architecture. Two distinct polypeptide stretches (total of 77 residues) from the LC variable domain ($V_l$) fit the fibril density. Despite $V_l$ high sequence variability, residues stabilizing the fibril core are conserved through different cardiotoxic $V_l$, highlighting structural motifs that may be common to misfolding-prone LCs. Our data shed light on the architecture of LC amyloids, correlate amino acid sequences with fibril assembly, providing the grounds for development of innovative medicines.

[1] Dipartimento di Bioscienze, Università degli Studi di Milano, Via Celoria 26, 20133 Milano, Italy. [2] Centro di Ricerca Pediatrica Romeo ed Enrica Invernizzi, Università degli Studi di Milano, Via Celoria 26, 20133 Milano, Italy. [3] Amyloidosis Research and Treatment Center, Fondazione IRCCS Policlinico San Matteo, and Department of Molecular Medicine, University of Pavia, P.le Golgi 19, 27100 Pavia, Italy. [4] Department of Morphological and Physiological Sciences, Graduate School of Health Sciences,, Kumamoto University, 4-24-1 Kuhonji, Kumamoto 862-0976, Japan. [5] Department of Neurology, Graduate School of Medical Sciences, 1-1-1, Honjo, Kumamoto 860-0811, Japan. [6] Institute for Biomedical Technologies-CNR, Via Fratelli Cervi 93, 20090 Segrate, Italy. These authors contributed equally: Paolo Swuec, Francesca Lavatelli. Correspondence and requests for materials should be addressed to S.R. (email: stefano.ricagno@unimi.it) or to M.B. (email: martino.bolognesi@unimi.it)

Light chain amyloidosis (AL), with an incidence of about 10 new cases per million-persons/year, is currently the most common systemic form of amyloidosis in Western countries[1]. The disease is associated with the presence of a plasma cell clone, and is caused by extracellular deposition of misfolding-prone monoclonal immunoglobulin light chains (LC), transported to target organs through blood. Deposition of amyloid fibrils is associated with dysfunction of affected organs. The amino acid sequence of each patient's monoclonal LC is virtually unique, as a consequence of immunoglobulin germline genes rearrangement and somatic hypermutation. Fibril deposition in AL is widespread, and can target different organs; heart involvement dramatically worsens patients' prognosis[2–4]. Much research is currently being devoted to defining the molecular bases of amyloid cardiomyopathy[5–7], to hinder fibrillogenesis[8] and cell damage[5,9,10].

LC subunits (ca. 215 residues) consist of two β-sandwich domains, each hosting a disulfide bridge: the highly variable N-terminal domain ($V_l$, ca. 105 residues), a short joining region ($J_l$), and the C-terminal constant domain ($C_l$)[6,11]. Both full-length LCs and isolated $V_l$ domains are typical components of the deposited fibrils;[12,13] nonetheless, the mechanisms promoting aggregation in vivo remain unclear. Progress in understanding LC aggregation is hampered by lack of structural insight on AL fibrils, only low-resolution characterization of LC fibrils being available to date[14,15].

Cryo-EM is currently the first-choice method for the structural analysis of amyloids[16–20]. Notably, in the few studies reported to date, the protein hosted within the fibril was shown to adopt composite folds, compatible with, but not fully predictable from, fibril models based on short peptides[21]. Moreover, whether samples prepared in vitro or in model systems truly represent the fibril structures accumulated in patients remains an open question. Recent structural work on Tau protein fibrils well demonstrated that the same polypeptide chain can assume different folds within the fibrils[16,17], and that in vitro grown fibrils may not recapitulate the structural features observed in patient deposits[22]. One further question concerns systemic amyloidosis, where the involved amyloidogenic proteins are typically natively folded under physiologic conditions. It is in fact unclear whether natively folded proteins need to unfold completely before re-assembly into cross-β structure, but also whether the native and fibril folds should bear any structural resemblance. Thus, it can be argued that structures of fibrils grown under denaturing conditions, and perhaps in animal models, may not completely address features present in patients' amyloids[20,23].

The above considerations prompted us to focus our studies on the characterization of patient-derived amyloid fibrils. Here we present the cryo-EM structure, at 4.0 Å overall resolution, of ex vivo LC fibrils extracted from the heart of a patient affected by severe AL cardiac amyloidosis. We show that the ex vivo fibrils are composed of an asymmetric protofilament hosting 77 residues from the LC $V_l$ domain, coupled to two low-order regions that comprise about one-third of the $V_l$ domain and portions of the $C_l$ domain. Consideration of proteolytic patterns, fibril structural motifs, and of amino acid sequences suggests mechanisms for aggregation and fibril elongation in AL amyloidosis.

## Results and Discussion

### Characterization of amyloid deposits in AL amyloidosis.
In order to explore the structural organization of natural amyloid fibrils, we extracted and characterized ex vivo amyloid aggregates from the affected heart tissue. Specifically, fibrils were isolated from left ventricle specimens acquired during autopsy from a male patient affected by AL λ amyloidosis, with severe amyloid cardiomyopathy. Microscopic analysis of cardiac tissue showed extensive extracellular amyloid accumulation (Fig. 1a, b). The monoclonal amyloidogenic LC responsible for such deposits, labeled AL55, was sequenced from its coding mRNA from bone marrow plasma cells; AL55 is of λ isotype and belongs to the *IGLV6-57* germline gene, which is overrepresented in the repertoire of amyloidogenic LCs, compared to the polyclonal repertoire[24,25].

The molecular composition of the isolated fibrils was analyzed through a proteomic approach, based on two-dimensional polyacrylamide gel electrophoresis (2D-PAGE) followed by nano-liquid chromatography tandem mass spectrometry (nLC-MS/MS) analysis of excised protein spots. In agreement with previous observations[12,13], AL55 fibrils are composed of a heterogeneous population of LC proteoforms and N-terminal LC fragments (Fig. 1c, Supplementary Figure 1). Besides species corresponding to the intact AL55, as commonly observed in AL λ deposits[13], chain fragments whose molecular weight is lower than the full-length LC are predominant in the fibrils; all host a complete $V_l$ domain and extend through regions of the $C_l$ domain. After tryptic digestion of the protein spots, our nLC-MS/MS data allowed to identify AL55 fragments extending to residues 129, 134, or 150, i.e. to residues located at the distal ends of the first or of the third β-strands in the $C_l$ domain, respectively (Supplementary Figure 1d–f). Low molecular weight fragments are resistant to fibril limited proteolysis in vitro (Supplementary Figure 1c); in contrast, degradation of the longer proteoforms is observed, suggesting protection or burial of the $V_l$ domain within the assembled fibril.

### Cryo-EM structure of the AL55 fibrils.
Contrary to previous reports[14], our negative staining electron microscopy and cryo-EM analyses of freshly extracted AL55 amyloid material revealed no polymorphs (Fig. 1d, Supplementary Figure 2). AL55 fibrils display a helical pitch of $1070 \pm 30$ Å and 80–175 Å width range (Supplementary Figure 2). Absence of two symmetric protofilaments is evident from inspection of the fibrils in raw micrographs, and in images obtained by reference-free 2D classification of segments comprising an entire helical pitch (Fig. 1d, Supplementary Figure 2). 2D class averages of vitrified AL55 fibrils, where β-strands are clearly resolved, suggest the presence of a highly ordered core surrounded by low-order regions (Fig. 1e, Supplementary Figure 2). Cryo-EM 3D reconstruction of AL55 fibrils resulted in a map at overall resolution of 4.0 Å, in which the cross-β structure was clearly resolved (Fig. 2, Supplementary Figure 3, Supplementary Table 2). Consistent with the 2D analyses, the AL55 density map showed a fibril core whose consecutive β-strand rungs are related by helical symmetry, with a rise of 4.9 Å, a twist of approximately –1.6°, and two low-order outer regions.

For each LC subunit deposited along the fibril axis, the inner structured core is divided into two segments: the central part of the density displays a "snail-shell" trace that is surrounded by a second, "C-shaped", extended polypeptide stretch (Fig. 2a). The two regions are spatially contiguous but not directly connected by interpretable density, indicating that two distinct LC segments build the fibril core. Several bulky side-chains visible in the map, together with the Cys22–Cys91 disulfide bridge, supported chain tracing and modeling for 77 residues of AL55 $V_l$. As a result, the first N-terminal 37 residues map into the internal snail-shell region, while the outer C-shaped stretch hosts residues 66–105 (Fig. 2d). Individual LC subunits assemble with a parallel β-sheet topology along the fibril elongation axis, i.e. along the inter-subunit H-bonding direction. Each subunit presents nine β-strands; β1–β5 belong to the snail-shell region, and β6–β9 pack

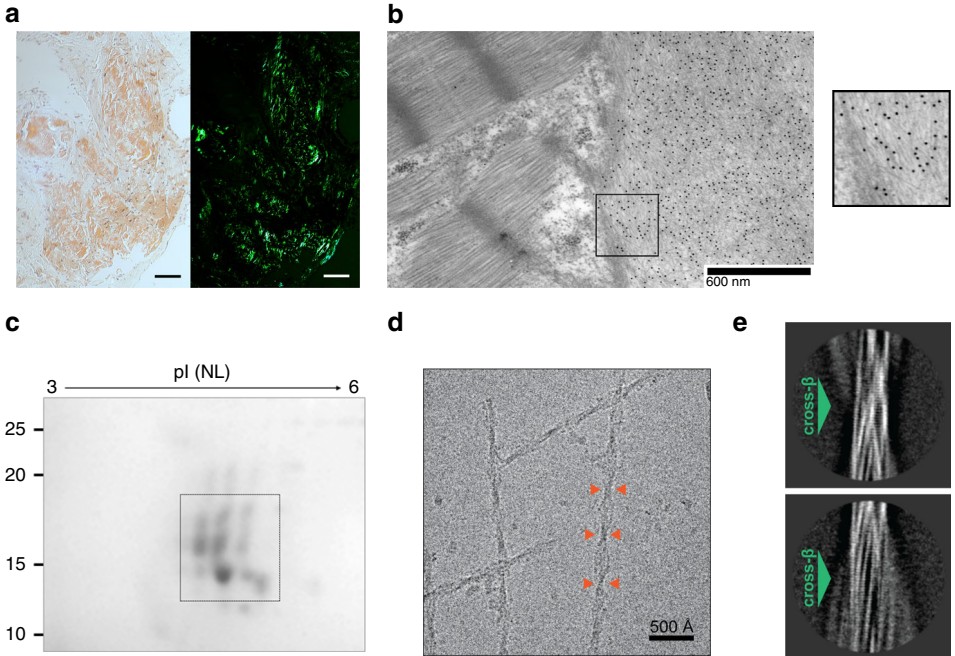

**Fig. 1** Morphological and molecular characterization of AL55 fibril deposits. **a** Myocardial tissue from patient AL55, stained with Congo red. Red-orange stain and apple-green birefringence indicate amyloid deposits under visible (left) and under polarized light (right), respectively (magnification ×100; scale bar 100 μm). **b** Immuno-electron microscopy imaging of heart tissue from patient AL55 (magnification ×6000). Extracellular amyloid fibrils are visible on the right side: the gold-conjugated secondary antibody appears as black dots. The cardiomyocyte sarcomeres are visible in the upper left corner. Scale bar: 600 nm. The squared portion is zoomed in the inset on the right. **c** 2D-PAGE analysis of purified AL55 LC fibrils (inset from Supplementary Information Fig. 1b, left panel), showing the spots identified by MS as AL55 LC fragments (framed). These encompass the full $V_1$ and variable portions of $C_1$ (MS sequence coverage ranges from aa 1–129 to aa 1–150). Low and high MW fragments comprise residues from the N-terminus to residue 129, and to 150, respectively. **d** Representative cryo-EM micrograph of AL55 LC fibrils; orange arrows highlight fibril cross-overs. **e** Reference-free 2D class averages of AL55 fibril showing distinct cross-β staggering (green arrows)

around it in the C-shaped stretch (Fig. 2f). In particular, strands β1, β3, β5, and β6 face each other and tightly pack their side chains together, while β4, β7, and β9 form a second contact region of lower side chain packing density.

As previously reported for other amyloid fibril structures[16–18,20,26], the β1–β5 modeled strands do not lay in a planar arrangement within the fibril; rather, the snail-shell region adopts a β-helix-like structure. In particular, the polypeptide chain of LC subunit $i$, whose N-terminus rests on plane $i$, rises twice along the fibril axis, at residues Pro14 (to $i+1$ level) and at Trp36 (to $i+2$ level). As a result, and looking along the opposite fibril axis direction, the side chains from subunit $i$ strand β1 pack against the C-terminal residues of subunit $i-2$ (Fig. 3a). On the contrary, the C-shaped region (β6–β9) lays essentially in a plane; given that it is covalently bound to the snail-shell region through the Cys22–Cys91 disulfide bond, this segment in subunit $i$ is located at the $i+1$ level (Fig. 3a). Such overall assembly produces fibril ends that are not flat. Thus, analogously to what has been discussed for amyloid-β (Aβ1–42) fibrils[18], the two fibril ends present a groove and a ridge (Fig. 3b, c), both of which expose highly hydrophobic patches (β1–β3 interface). Conceivably, the edge β1-strand of a natively folded LC could be recruited through interaction with the hydrophobic groove/ridge of a growing fibril, promoting unfolding and association of a new subunit.

**AL55 unfolding is an obligate step for amyloid formation.** Figure 4 compares a molecular model of the natively folded AL55 $V_1$, displaying the typical immunoglobulin β-sandwich domain, with the fibril structure of AL55 $V_1$ (Fig. 4a, c), and allows a few general considerations. Native AL55 $V_1$ displays a compact domain composed of two antiparallel β-sheets; conversely, in

fibrils, 77 $V_1$ residues adopt a relatively planar organization devoid of intramolecular β-sheet-like interactions. The natively folded $V_1$ domain is fully structured for ca.105 residues, while about one-third of the fibril $V_1$ (residues 38–65) fall in a poorly structured region. Moreover, although both the native and the fibril structures essentially consist of β-strands and loops, the locations, number, and extensions of such structures are markedly different in the two states (Fig. 4b). Essentially none of the side-chain interactions present in the tertiary structure of native AL55 are conserved in the fibril. Two salt bridges stabilize the fibril core: expectedly, Lys82–Glu84 is also present the native folded state, while the Lys16–Asp95 salt bridge links residues that are 40 Å apart in the folded AL55 $V_1$ domain. Similar considerations hold for hydrophobic contacts: none of the hydrophobic contacts observed in the fibril (Phe2–Ile20; Leu4–Leu18, and Ile29–Leu81; Ile29–Ile78 and Ala30–Leu76) can be achieved in the domain folded state. The above observations suggest that in vivo major unfolding of AL55 $V_1$ is an obligate step for amyloid formation.

**Role of the AL55 sequence in fibril assembly.** Overall, the AL55 segments that build the structured fibril core are characterized by a few charged residues (9 out of 77), four of which are involved in two salt bridges (Lys16–Asp95, Lys82–Glu84) alleviating Coulombic repulsion (Fig. 5a, b). The 82–88 segment, which comprises most of the charged residues, is detached from the inner fibril core (Fig. 5b) and may prove accessible to water molecules. The $V_1$ stretches predicted to be most aggregation prone[27] are all comprised in the fibril core (Fig. 4b) that hosts two hydrophobic clusters. The first cluster builds the β1–β3 interface (Phe2, Leu4, Leu18, Ile20), while the second is located between

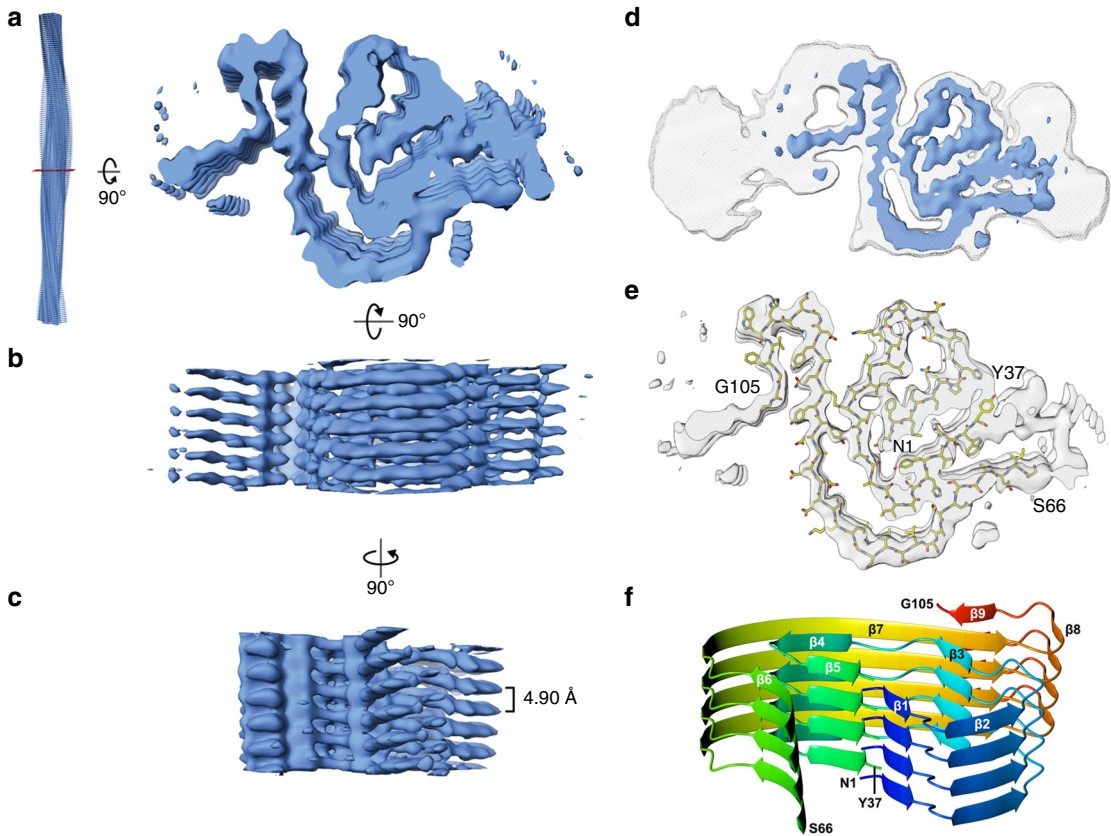

**Fig. 2** Structure of AL55 amyloid fibrils. **a–c** Orthogonal views of the post-processed 3D reconstruction at overall resolution of 4.0 Å (FSC = 0.143). **d** Overview of the AL55 fibril cryo-EM map covering the structured fibril core region (blue). Cross-sectional EM densities of sharpened, 4.0 Å (blue) and unsharpened, 4.5 Å low-pass filtered (gray) reconstructions. **e** Atomic model of AL55 (residues 1–37 and 66–105) superimposed on a cross-section of the EM density map. **f** Ribbon representation of the fibril structured core, rainbow colored. Four stacks (subunits) in the typical cross-β arrangement are shown

the β4–β5 turn and β6–β7 (Ile29, Leu76, Ile78, Leu81) (Fig. 5a), and includes the disulfide bond linking β4 to β7. The two prolyl residues in the fibril core (Pro7 and Pro14) help forming the β1–β2 and β2–β3 turns, respectively. In the native $V_1$ immunoglobulin fold Pro7 is located at the center of the edge β1-strand that is bulged; notably, such bulged edge strands are held to protect against amyloid aggregation[28]. Nevertheless, in the AL55 fibril core Pro7 conveniently accommodates in the β1–β2 turn.

Besides the structured core region, as mentioned above (Fig. 1e, and Supplementary Figure 2), AL55 fibrils present two areas of lower order: the first falls between residues 37 and 66 of the structured segments (Fig. 2). As limited proteolysis suggests that the whole $V_1$ domain is protected in the fibril core, we postulate that the 38–65 polypeptide fills the low-order region adjacent to residues 37 and 66. The 38–65 sequence hosts residues with low β propensity (four Pro and three Gly) and seven charged residues, which might prevent the formation of regular β-structures once out of the folded immunoglobulin domain context. The second poorly structured zone surrounds the last core modeled residue (Gly105) (Fig. 2d). nLC-MS/MS analysis shows that the fibrils contain portions of the $C_1$ domain (that follow beyond Gly105 in the intact LC). Similar to the 38–65 low-order region described above, the portion of AL55 downstream of the three consecutive Gly residues terminating the structured fibril core hosts residues with low aggregation propensity, such as four Pro and four charged residues that may trigger the onset of this low-order region. Moreover, heterogeneity in extension of the AL55 C-terminal region can also contribute to packing defects and conformational variability within the fibril, thus negatively affecting the local cryo-EM density observed (Fig. 2d).

**LC sequence and fibril assembly.** AL55 sequence belongs to the λ6 subgroup, and in particular to the *IGLV6-57* germline gene, which is expressed in about 2% of in bone marrow plasma cells expressing λ LC[25], but makes up to 18% of all the monoclonal λ proteins responsible for AL amyloidosis[24]. Given the relevance of such subgroup in AL pathogenesis, several previous studies have analyzed proteins belonging to the *IGLV6–57* segment, their aggregation propensity and the role of specific positions in tuning protein stability and amyloidogenicity[29]. In particular, the interaction between residues Phe2 and Arg25 highly stabilizes the $V_1$ native fold, while amino acid variations in one of the two positions increases the overall aggregation propensity[30,31]. Gly at site 25, as occurring in AL55, is reported to facilitate amyloid formation[31].

The LC fibril model, here presented, shows that two AL55 hypervariable complementarity determining regions (CDR) contribute to the structured fibril core. CDR1 (Thr23-Gln35) spans strands β3 through β4; CDR3 (Gln92-Val101) falls at the C-end of the modeled fibril core. Such observations highlight sequence variability of $V_1$ domains as a key factor not only for protein instability and aggregation propensity, but also for the molecular interactions that stabilize fibril assembly. Thus, questions arise on how common the fibril architecture here reported might be for AL deposits due to LCs other than AL55. Structure-based considerations can be drawn.

Firstly, AL55 solubilized fibrils provide a typical pattern in 2D-PAGE observed in the other solubilized AL deposits from patients[13]. The fragments of molecular weight lower than the full-length LC are highly abundant and consist of the $V_1$ domain, with or without a short stretch of $C_1$ domain; even the shorter

fragments always contain the LC N-terminal region[13], strongly suggesting that the $V_1$ domain is constantly required to build the fibril core. To assess whether other LCs may be compatible with the fibril architecture here reported, AL55 sequence was aligned

with eight diverse cardiotoxic λ LCs (Fig. 5c) previously reported[6]. Despite sequence variability, the residues that appear to play structuring roles in AL55 fibrils are conserved or conservatively mutated. An N-terminal stretch of hydrophobic residues, building the fibril inner β1-strand, is present in all sequences (Fig. 5a, c). Frequently, an extra prolyl residue (Pro8) can be found, which would be located in the β1–β2 turn, not impairing the fibril fold observed for AL55. Indeed, previous evidence showed that the His8→Pro mutant in the $V_1$ domain belonging to the *IGLV6–57* slowed but did not abrogate fibril formation[32]. Gly15, which in the AL55 fibril core adopts a conformation unfavorable for other amino acids, is conserved in all sequences. The conserved disulfide bond is a strong structural restrain. The two hydrophobic clusters stabilizing the AL55 fibrils can be assembled in all other eight λ LCs. Moreover, several aromatic residues that stabilize the fibril through inter-subunit stacking interactions, and the (82–88) segment comprising charged residues, are present in all sequences (Fig. 5a, c). A recent ssNMR model of in vitro fibrils formed by a $V_1$ belonging to the *IGLV6–57* gene segment and with only 12 mutations when compared to AL55 sequence, shows several structural analogies. Even though the N-terminal stretch is predicted partially flexible, both the reported polymorphs display two ordered regions (residues 20–45 and 65–103) with parallel arrangement and a disordered region spanning residues 45–60 (ref. [33]).

Hence, overall, while the sequence variability observed in $V_1$ domains might result in different fibril structural arrangements, as observed for example for different isoforms of the Tau protein[16,17], nevertheless, our results strongly suggest that the structural motifs observed in the AL55 fibril architecture are compatible with the assembly of amyloid deposit from different LCs, opening the way to an innovative and targeted molecular characterizations of AL amyloidosis.

Finally, some considerations on the role of proteolysis in AL amyloidosis may be drawn. Although the presence of multiple N-terminal LC fragments is a universal finding in AL amyloid fibrils, it is however unclear whether proteolysis releases amyloidogenic LC fragments, which then assemble into fibril deposits, or whether proteolysis occurs after amyloid formation. Recent reports suggest that susceptibility to proteolysis is distinctive for amyloidogenic LCs[6,34,35]. Our nLC-MS/MS analysis of AL55 fibrils allowed the identification of peptides from LC fragments extending to the distal ends of the first or of the third $C_1$ domain strands. These protein regions are solvent

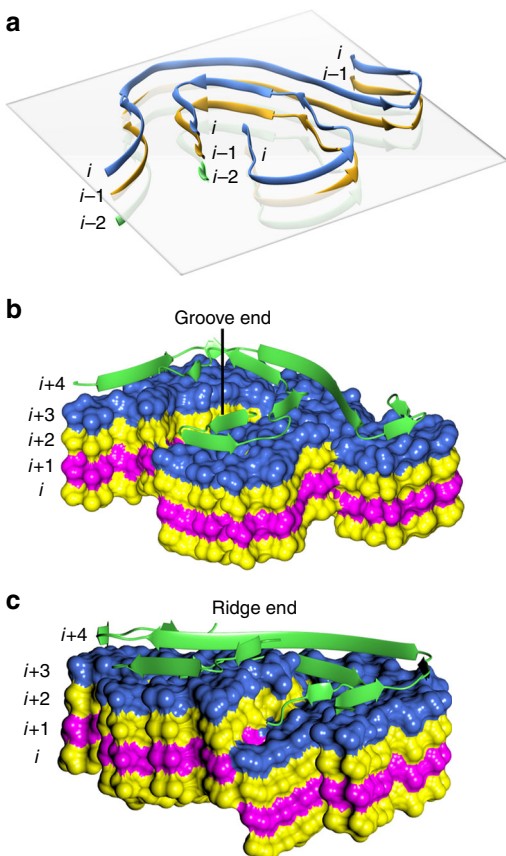

**Fig. 3** Fibril subunits are not planar. **a** A horizontal plane (perpendicular to the fibril elongation axis) is added to this ribbon representation of the fibril core to highlight the raise of each subunit at the end of the β2 strand (Pro15) and of β5 (Trp36), respectively. **b, c** A structural view of the fibril ends: the inner core concave region (groove) is shown in **b**; the inner core (β1–β2) of the fibril is shown in **c** protruding from the surface (ridge). Different colors are used for independent subunits

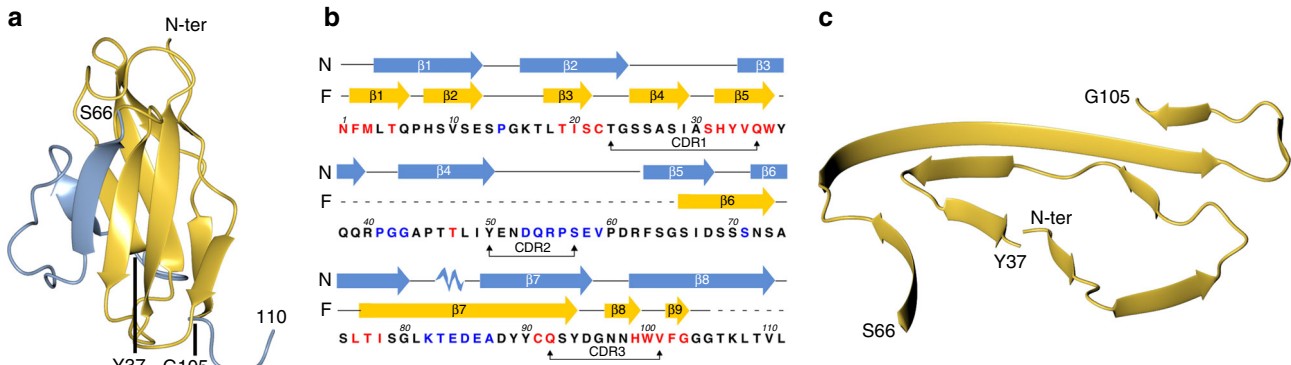

**Fig. 4** Comparison of native and fibril AL55 structures. **a** Ribbon representation of AL55 $V_1$ domain native structure (model). The ordered and disordered regions in the structure of the fibril core are in yellow and blue, respectively. **b** Sequence of AL55 $V_1$ domain: scheme of the secondary structure elements of the natively folded LC (N, blue), and of the fibril assembly (F, yellow), respectively; dashed lines correspond to residues not-modeled in the EM density. Residues are colored according to their intrinsic aggregation propensities, as defined by CamSol[27] (high, in red; low, in black; very low in blue). **c** Ribbon representation of the AL55 $V_1$ domain fibril structure

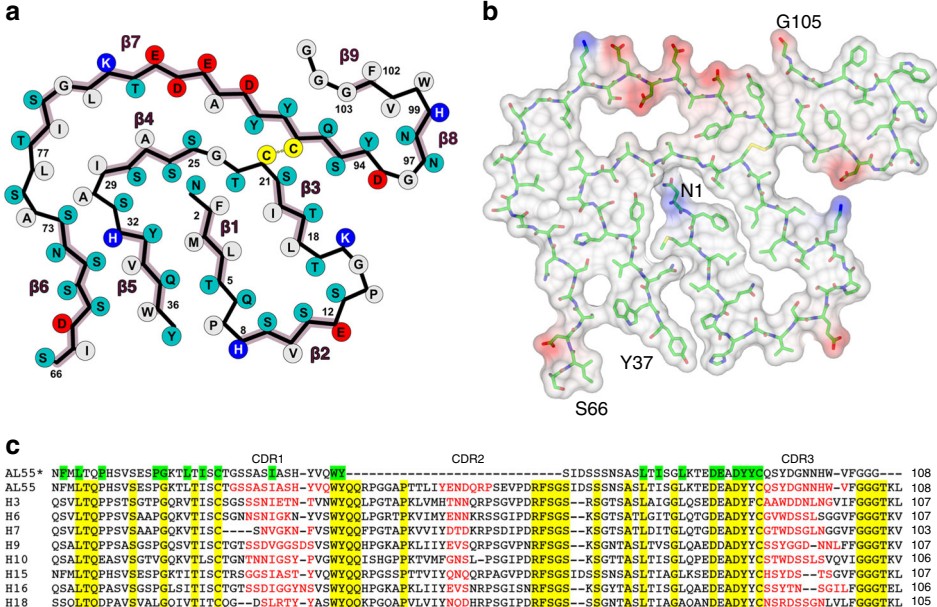

**Fig. 5** The fibril assembly depends on AL55 sequence. **a** A 2D schematic view of the core polypeptide stretches. Residues are colored as follows: gray hydrophobic, cyan polar, red and blue negatively and positively charged, respectively. **b** Stick representation of the fibril core together with a surface representation colored according to electrostatic charges. **c** AL55 $V_l$ is aligned against eight previously reported LCs responsible for severe cardiac amyloidosis; such LCs display highly diversified sequences and belong to distinct germlines[6]. The first line shows only AL55 residues modeled in the map. Residues considered particularly relevant in determining AL55 fibril structured core fold are highlighted in green. Conserved residues are highlighted in yellow. The nine LCs here aligned belong to the following germlines: AL55 *IGLV6–57*; H3 *IGLV1–44*; H6 *IGLV1–51*; H7 *IGLV1–51*; H9 *IGLV2-8*; H10 *IGLV1–36*; H15 *IGLV6–57*; H16 *IGLV2–14*; H18 *IGLV3–19*

exposed in the native LC domain structure (Supplementary Figure 1), thus proteolysis could feasibly take place when the LC chains are natively folded. In turn, such cleavages may well destabilize (i.e. start unfolding of) the $C_l$ domain, whose structural integrity is known to play a stabilizing role for the full LCs[36]. The overall structure of the AL55 fibrils on the other hand, shows that the $C_l$ domain is not protected in the mature fibrils, thus it might be completely removed by proteolysis occurring on fibril deposits. Taken together, the above structural and biophysical considerations allow speculating that LC proteolysis may occur to a large extent before aggregation.

In summary, the data here reported present a cryo-EM characterization of ex vivo fibrils from a patient affected by the most common systemic form of amyloidosis. Together with the recent work on the Tau protein[16,17], to date, this is the only structural analysis of fibrils carried over on materials directly extracted from human tissue under pathologic conditions. The data and associated considerations help shedding light on some of the basic molecular aspects of AL amyloidosis and, more in general, of systemic amyloidoses. AL55 fibrils present an asymmetric single protofilament; 77 residues belonging exclusively to the AL55 $V_l$ domain are required to build the structured core, and to extend the fibril through intermolecular hydrogen bonds and hydrophobic contacts. Despite the high sequence variability typical of $V_l$ domains, the amino acid sequences of other amyloidogenic (cardiotoxic) LCs are compatible with the fibril main structural motifs here identified. Having shed light on the structure of ex vivo fibrils that result from an otherwise natively folded protein, our results, as a whole, support the concept that LC unfolding is needed along the aggregation pathway in AL disease. In this context, we propose a major role for proteolysis in triggering the LC amyloid aggregation process.

## Methods

**Fibril extraction from heart**. Heart tissue (left ventricle) was acquired during autopsy from a patient (herein indicated as patient AL55) affected by systemic AL λ amyloidosis, whose autopsy examination showed Congo red positivity in heart, kidney, lung, liver, pancreas, and spleen. The patient died of acute bronchopneumonia with alveolar damage. Tissue was stored frozen (−80 °C) without fixation until use. The diagnosis of amyloidosis had been made 4 years earlier; the main clinical features are described in Supplementary Table 1. The patient was treated with oral melphalan and dexamethasone[37–39], with achievement of complete hematological remission and organ response. Six months before the patient's death, reappearance of the serum monoclonal component was documented, but no therapy was started due to the stability of organ damage. Bone marrow had been withdrawn from the same patient during the routine diagnostic procedures, upon acquisition of the informed consent for storage and use of samples for research purpose. This study has been approved by the Ethical Committee of Fondazione IRCCS Policlinico San Matteo and was performed in accordance with the Declaration of Helsinki. The presence of amyloid deposits was evaluated by Congo red staining analysis under polarized light and by electron microscopy. Amyloid typing was confirmed by immuno-electron microscopy[40]. Organ involvement was defined according to international criteria[41]. Baseline clinical and demographic information has been collected (Extended Data Table 1). Fibrils were extracted as described by Annamalai et al.[42]. Briefly, 0.5 g of tissue were cut in small pieces and repeatedly washed in Tris calcium buffer (20 mM Tris, 140 mM NaCl, 2 mM CaCl₂, pH 8.0; each washing step was followed by centrifugation at 3100*g* at 4 °C). Tissue was then digested overnight with *Clostridium histolyticum* collagenase (Sigma Aldrich, Saint Louis, MO, USA) (5 mg/ml in Tris calcium buffer at 37 °C). The material was centrifuged at 3100*g* for 30 min at 4 °C. The pellet was manually homogenized with a glass Potter pellet pestle in 1 ml of Tris EDTA buffer (20 mM Tris, 140 mM NaCl, 10 mM EDTA, pH 8.0). The homogenate was centrifuged for 5 min at 3100*g* at 4 °C. This step was performed for 10 times overall. The remaining tissue pellet was homogenized with a glass Potter pestle in 1 ml of ice-cold water. The homogenate was centrifuged for 5 min at 3100*g* at 4 °C and the supernatant was stored. This step was repeated six more times, without pooling supernatants. The cryo-EM and proteomic analyses were performed on fraction 3. Immediately after protocol completion complete protease inhibitor cocktail (Roche, Basel, Switzerland) was added to fibrils (with the exception of fibrils destined for limited proteolysis experiments), and the sample tubes were stored on ice until analysis.

**cDNA sequencing of monoclonal free LC**. Total RNA was extracted from $10^7$ bone marrow mononuclear cells with TRIzol reagent (Life Technologies, Paisley, United Kingdom). Monoclonal λ LC sequence was cloned by a universal inverse-

PCR strategy[43], using primers specific for the 5′ (5′-AGTGTGGCCTTGTTGGCT TG-3′) and 3′ (5′-GTCACGCATGAAGGGAGCAC-3′) portions of the λ LC constant region. The germline gene of V region was deduced by sequence alignment with the current releases of EMBL-GenBank (https://www.ncbi.nlm.nih.gov/genbank/), V-BASE (http://www.vbase2.org/) and IMGT (http://www.imgt.org/) sequence directories. In order to obtain the original full-length LC sequence, standard RT-PCR was employed using the same RNA, N-terminal monoclonal V region primer (5′-AATTTTATGCTGACTCAGCCC-3′) and a universal Cλ carboxyterminal primer, corresponding to the last amino acids of the constant region (5′-TGAACATTCTGTAGGGGCCAC-3′)[44]. After purification of recombinant plasmid, insert was sequenced from both sides.

**Limited proteolysis**. Proteins were quantified using BioRad Protein assay (BioRad, Hercules, CA, USA) after 1:4 dilution of the fibril suspension with isoelectrofocusing (IEF) buffer (7 M urea, 2 M thiourea, 4% CHAPS, 0.1 M DTT) and incubation for 30 min at room temperature, in order to disrupt the aggregates. For limited proteolysis, a volume of purified fibrils corresponding to 20 μg of proteins was incubated with 0.1 μg/ml of Proteinase K (Sigma Aldrich, Saint Louis, MO, USA), at 37 °C, in a buffer containing 10 mM Tris pH 8, 10 mM NaCl, 5 mM CaCl₂. The reaction was stopped after 1 h by dilution 1:4 (v:v) with a solution containing 7 M urea, 2 M thiourea, 4% CHAPS, 0.1 M DTT, and freezing at −80 °C.

**Two-dimensional polyacrylamide gel electrophoresis**. For 2D-PAGE analysis, 25 μg of fibril proteins were diluted (1:4 v:v) in IEF buffer, with addition of pI 3–10 ampholytes (Bio-Rad) to a final concentration 0.02% v/v (final volume ~200 μl) and incubated for 30 min at room temperature. First and second electrophoretic dimensions were performed using, respectively, 11 cm strips, non-linear 3–10 pH gradient (Bio-Rad) and 8–16% polyacrylamide gradient midi format gels (Criterion TGX gels, Bio-Rad). IPG strips were subjected to active rehydration at 50 V for 12 h. Isoelectric focusing was performed in a Bio-Rad Protean™ IEF cell as follows: 250 V for 15 min, increase up to 8000 V over 1 h and 8000 V steady until a total of 35,000 V–h had elapsed. Proteins were reduced and alkylated (using iodoacetamide) between first and second dimension. All gels were stained with colloidal Coomassie blue (Pierce, Thermo Fisher Scientific, Waltham, MA, USA) and imaged using a Gel Doc XR imaging system (Bio-Rad). For western blotting, proteins were transferred onto a PVDF membrane (Bio-Rad) using a trans Blot Turbo apparatus (Bio-Rad) and probed with polyclonal rabbit anti-human λ LCs (Dako, Agilent, Santa Clara, CA, USA) used at a concentration of 1 μg/ml, followed by incubation with an horseradish-peroxidase conjugated swine anti-rabbit secondary antibody (Dako).

**Protein spots analysis and identification**. Protein spot excision and in-gel digestion were performed as previously described[13]. Tryptic digests from each spot were analyzed using an Eksigent nanoLC-Ultra 2D System combined with cHiPLC-nanoflex system (trap-elute mode) (Eksigent, AB SCIEX Dublin, CA, USA) coupled to Q Exactive mass spectrometer (Thermo Fisher Scientific, San Josè, CA, USA). Briefly, tryptic digests were first loaded on the cHiPLC trap (200 μm × 500 μm ChromXP C18-CL, 3 μm, 120 Å) and washed in isocratic mode with 0.1% aqueous formic acid for 10 min at a flow rate of 3 μl/min. The automatic switching of cHiPLC ten-port valve then eluted the peptide mixture on a nano cHiPLC column (75 μm × 15 cm ChromXP C18-CL, 3 μm, 120 Å), through a 40 min gradient of 5–60% acetonitrile (containing 0.1% formic acid), at a flow rate of 300 nl/min. Trap and column were maintained at 35 °C. A Q Exactive mass spectrometer was equipped with a nanospray ionization source using a coated fused silica emitter (New Objective, Woburn, MA, USA) (360 μm o.d./50 μm i.d.; 730 μm tip i.d.) held at 1.9 kV. The ion transfer capillary was held at 220 °C. Full mass spectra were acquired in positive ion mode over a 400–1600 $m/z$ range and with a resolution setting of 70000 FWHM (@ $m/z$ 200) with 1 microscan per second. Each full scan was followed by seven MS/MS events, acquired at a resolution of 17,500 FWHM, sequentially generated in a data-dependent manner on the top seven most abundant isotope patterns with charge ≥2, selected with an isolation window of 2 $m/z$, fragmented by higher energy collisional dissociation (HCD) with normalized collision energies of 30 and dynamically excluded for 30 s. Data were processed using the Sequest HT-based search engine contained in the Thermo Scientific Proteome Discoverer software, version 2.1. using a human protein database downloaded in January 2018 from UNIPROT, and augmented with the sequence of AL55. The following criteria were used for the identification of peptide sequences and related proteins: minimum precursor mass 400 Da, maximum precursor mass 5000 Da (S/N ratio for peak filter 1.5); maximum missed cleavage per peptide 3; minimum peptide length 6 amino acids; maximum peptide length 144 amino acids; tolerance on precursor mass was set at 10 ppm and on fragment mass at 0.05 Da. Percolator (maximum delta Cn 0.05 and maximum rank 1). Target false discovery rate was 0.01 in strict mode. Validation was based on $q$-value.

**Electron microscopy sample preparation**. Freshly extracted AL55 fibrils were first analyzed by negative staining EM. Briefly, a 4-μl droplet of sample was applied onto a 400-mesh copper carbon-coated grids (Agar Scientific), glow discharged for 30 s at 30 mA using a GloQube system (Quorum Technologies). After 1-min

incubation, excess of sample was removed and the grid was stained with 2% (wt/v) uranyl acetate solution, blotted dry, and imaged on a LEO 912Ab transmission electron microscope (Zeiss) operating at 100 keV. For cryo-EM grid preparation, a 3-μl droplet of freshly extracted AL55 fibrils was applied onto a glow discharged holey carbon grids (Quantifoil R1.2/1.3, 300-mesh), incubated for 30 s, and plunge-frozen in liquid ethane using a Vitrobot Mk IV (Thermo Fischer Scientific) operating at 4 °C and 100% RH.

**Cryo-EM data collection and image processing**. In total, 1680 images of vitrified AL55 fibrils were acquired on a Falcon 3EC direct electron detector (Thermo Fischer Scientific) using a Thermo Fischer Talos Arctica transmission electron microscope operating at 200 kV. Each image was acquired with an exposure time of 1 s and a total accumulated dose of 50 electrons per Å² equally distributed over 39 movie frames. Images were acquired at a nominal magnification of ×120,000, corresponding to a pixel size of 0.889 Å/pixel at the specimen level, with applied defocus values between −0.5 and −2.5 μm.

Prior to image processing, anisotropic magnification distortion was automatically estimated using mag_distortion_estimate[45]. Images were corrected for anisotropic magnification distortion (resulting in a corrected pixel size of 0.887 Å), motion-corrected and dose-weighted using MOTIONCOR2 (ref. [46]). Contrast transfer function (CTF) estimation was performed on aligned, unweighted sum images using CTFFIND4 (ref. [47]). Micrographs reporting resolution estimate of 5 Å or better were selected for further analysis. A total of 678 selected micrographs were imported in RELION 2.1 (refs. [48,49]) for subsequent image-processing tasks. Filaments were manually picked using RELION's helix picker from non-dose-weighted images. Segments were successively extracted using a box size of 320 pixel and inter-box distance of ~10% (28.2 Å), yielding a total of 104,689 segments.

An initial reference-free 2D classification was performed with regularization value of $T = 2$ to remove segments containing filament's termini and contaminants. Subsequent rounds of reference-free 2D classification were performed with regularization value of $T = 4$ to select for segments contributing to averages in which the filament's cross-β structure was clearly visible. A total of 97,723 segments were selected for subsequent 3D classification and refinement steps. Inspection of the squared amplitudes of the Fourier transform of the 2D class averages displaying the cross-β pattern showed a marked peak on the 1/(4.9 ± 0.05 Å) layer line. The initial estimate of the helical twist was calculated from measurements of AL55 filament crossover distances in cryo-EM images (1070 ± 30 Å).

An initial 3D model was obtained using relion_helix_toolbox. The model was low-pass filtered to 60 Å and used as reference for a 3D classification with single class ($K = 1$), regularization value of $T = 4$ and imposing a helical rise of 4.90 Å and a helical twist of −1.68°. The resulting 3D reconstruction was low-pass filtered to 30 Å and used as a reference for subsequent rounds of 3D classification using multiple classes ($K = 4$), a regularization values of $T = 10$ and local optimization of helical twist and raise parameters. For each round of classification, segments contributing to 3D class displaying β-strand separation along the helical axis were selected for subsequent reconstructions. A total of 21,031 segments were used for 3D auto-refine procedure using a 7 Å low-pass filtered map from previous 3D classification, a helical z_percentage parameter of 10%, and allowing the optimization of helical twist and rise. Finally, the refined 3D reconstruction was sharpened using RELION's standard post-processing procedure applying a soft-edge solvent mask and a β-factor of −106. The overall resolution estimate of the final map was 4.0 Å, calculated from Fourier shell correlations at 0.143. Helical symmetry was imposed on post-processed 3D map using the relion_helix_toolbox. Estimation of local resolution was performed using RELION 2.1. Further details are listed in Extended Data Table 2.

**Model building and refinement**. The initial model was prepared using Chimera[50] by assembling three-residues poly-Ala fragments into the final sharpened cryo-EM density map. The poly-Ala model was then refined using Coot and Phenix[51,52]. Subsequently, each Ala residue was mutated according to consensus between the AL55 protein sequence and final EM map features. Residues assignment was guided by the presence of an intramolecular disulfide bridge between Cys22 and Cys91 and other visible side-chain densities. The final model comprises AL55 residues 1–37 and 66–105. Real space refinement using Phenix, with a resolution cut-off of 4.0 Å, was performed on a fibril fragment consisting of five subunits (each one comprising the two segments 1–37 and 66–105). After each refinement stage, Coot was employed to manually adjust clashes, Ramachandran and rotamer outliers. During the refinements non-crystallographic symmetry constraints were imposed on the five subunits overall, and also on rotamers, C-beta deviations, Ramachandran plot, and secondary structure restraints. β-Sheet restraints were initially imposed on the whole sequence and then manually adjusted in later stages. In the last refinement stage, in addition to the gradient-driven model minimization, rigid-body refinement (where each segment of the five subunits was considered a rigid body) and grouped β-factor (ADP) refinement were employed, leading to a mean β-factor of 40.9 Å (ref. [2]). Molprobity was used for structure validation[53] and the EMRinger score[54] was calculated for a model comprising all the five subunits used in the refinement. Model building and refinement statistics are shown in the Extended Data Table 2. No Ramachandran outliers were detected and 91.8% of the residues were found in the favored regions of the Ramachandran plot. The residues falling in the allowed regions of the Ramachandran plot are Pro14, Thr17, Cys22,

Ala74, Ser79, and Gly80, which are mostly located at turns or kinks in the polypeptide. Secondary structures were analyzed using STRIDE[26]. Figures were prepared using Pymol, Chimera, and CCP4mg[50,55].

**Reporting summary**. Further information on experimental design is available in the Nature Research Reporting Summary linked to this article.

## Data availability

The AL55 cryo-EM map has been deposited with the Electron Microscopy Data Bank (code EMD-0274); the refined molecular model has been deposited in the Protein Data Bank (code 6HUD). Monoclonal LC sequence was deposited in GenBank database (accession number MH670901). Other data are available from the corresponding authors upon reasonable request.

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

## Acknowledgements

Contributions from the University of Milano and Fondazione Romeo and Enrica Invernizzi to founding the cryo-EM facility are gratefully acknowledged. This work was supported by Fondazione Cariplo (grants no. 2015-0591 and 2016-0489); Associazione Italiana per la Ricerca sul Cancro special program "5 per mille" (no. 9965); the Italian Ministry of Health (grants no. RF-2013-02355259 and RF-2016-02361756), Italian Medicines Agency (grant AIFA-2016-02364602), and E-Rare JTC 2016 grant ReDox. We are grateful to Laura Verga and Gianluca Capello for immuno-electron microscopy imaging.

## Author contributions

P.S., F.L., M.T., C.P., P. Milani, P.R., M.M. and F.B., performed the experiments; P.S., F.L. and S.R. designed the experiments; P.S., F.L., C.C., P. Mauri G.P., G.M., M.B. and S.R. wrote the paper with contributions from all other authors.

## Additional information

**Competing interests:** The authors declare no competing interests.

