## [Peer Review File · Nature Communications]

Reviewers' comments:

Reviewer #1 (Remarks to the Author):

The work by Swuec et al describes the 4.0 Å resolution structure of an amyloid fiber by CryoEM. This report is very relevant to the field of conformational diseases because the fiber was obtained ex vivo from the heart of a patient with amyloidosis.

The description of the structure is very clear and shows interesting features, in particular, the transformation of the native β barrel of the light chain into planar fibrils also formed by β strands.

My only concern with the article is the lack of reference and comparison with previous in vitro work about the folding and fibril formation of the protein encoded by the same germ line. The study of protein unfolding in vivo is very challenging, nevertheless there is abundant experimental information about the unfolding and refolding of this particular germline protein in vitro, for example, the authors state (p.11 end of first paragraph) that "energetic considerations suggest that such unfolding should not occur when the protein is in a monomeric state" in this context, the energetics (and kinetics) of the unfolding of the germline protein (Blancas-Mejia et al J.Mol.Biol. 2009), are relevant. Likewise, the authors describe the role of residues and regions of the protein, such as Pro7 and the N-terminal strand, whose role in stability and fibrillogenesis have been addressed in vitro (Hernández Santoyo et al J. Mol. Biol. 2010), (del Pozo Yauner et al Biochem.Biophys.Research.Comm. 2014). Such information is also available for proteins derived from patients (González-Andrade et al FEBS J 2013).

One of the more relevant points of the work by Swuec et al is that it shows a new structure with the adequate resolution to bring a step forward the transformation between the globular native fold and the amyloid fibril. In this context, a comparison of in vivo, ex vivo, in vitro and in silico approaches should be very fruitful. As stated by the authors, the differences between fibrils obtained from patients and those obtained in vitro or in model systems is an open question, therefore, a comparison with the NMR data recently reported for the fibrils formed in vitro for the same allotypic variant (Lecoq et al ChemBioChem 10.1002/cbic.201800732) should be relevant.

D. Alejandro Fernandez V.

Reviewer #2 (Remarks to the Author):

NCOMMS-18-37724-T - BOLOGNESI

Cryo-EM structure of cardiac amyloid fibrils from an immunoglobulin light chain (AL) amyloidosis patient

The paper by Swuec et al presents a 4.0 Å resolution cryo-EM structure of AL55 amyloid fibrils composed by two distinct polypeptide stretches in a total of 77 residues, which were extracted from a patient's heart. This work is novel as, in spite of the fact that there are currently a number of published structures of amyloid fibrils (amyloid-beta, tau, synuclein,...), this is one of the first reported structures for an AL amyloid fibril, and the field still needs to build up knowledge on this type of protein structures to better understand and mitigate its formation. Therefore, the information within this paper has a broader interest to the protein science and structural biology communities. The work is technically proficient, and the reported structural analysis is competent. The aspects below require some consideration:

a) A scale bar is missing in Fig 1a

b) In vivo unfolding is suggested to be an obligate step in fibril formation. At some point the authors state that "...energetic consideration suggest that unfolding should not occur ...in the monomeric state" (page 11, Lines 219-200). However, this needs to be either revised or substantiated with data as no evidence is provided to support this claim.

c) Subtitle 'AL55 unfolds along the aggregation pathway' is misleading in face of presented results in this paper, as the aggregation pathway is not studied. Paraphrasing the authors, this section should be changed to 'AL55 unfolding is an obligate step for amyloid formation' - p10, line 202
d) Proteolysis of AL55 as cause for amyloid formation stems from an inference from the author's data and is therefore essentially a working hypothesis/speculation. Therefore, this last section from results should be toned-down and a shorter version of this possibility can be rather (briefly) presented within the conclusions.

Reviewer #3 (Remarks to the Author):

The manuscript by Swuec et al. describes a cryo-EM structure of a fibril formed by immunoglobulin light chains (LC), which were extracted from the heart of a systemic light chain (AL) amyloidosis patient.

This work describes so far only the second structure of an amyloid fibril sample that has been extracted from a patient (following the work on the tau fibril by Fitzpatrick et al). This is already a great achievement and makes the structure highly interesting. Furthermore, the atomic structure of the AL fibril is an important step for understanding the molecular foundation of AL amyloidosis.

The structure determination (experiment and data analysis) seems sound and technically well done. The cryo-EM structure has a good resolution of 4.0 Ang and the density looks sufficiently well defined to allow for building an atomic model with reasonable confidence.

Interestingly, the fibril core contains a region of high sequence variability and the authors analyze convincingly how difference sequences could be accommodated by the presented fibril structure.

A point that is very interesting for the amyloid field in general is the fact that the LC is natively folded and the study shows that the protein has to completely refold to fold into the fibril structure, since none of the side-chain interactions are conserved and they differ in the fibril completely from those in the native structure.

The manuscript is very well written and results are clearly presented. I think the manuscript is very well suited for publication in Nature Communications.

Minor points:

- p.11: "while about one third of the fibril VI (residues 66-105) fall in a poorly structured region." The assignment "residue 66-105" seems to be a typo.

As shown further below, they probably refer to the residues 38-65 and around 105.

- p.11 "moreover, energetic considerations suggest that such unfolding should not occur when the protein is in a monomeric state but later on, along the aggregation pathway."

A reference or more explanation should be added, it is not clear to me what this means.

- Figure 5c "Conserved residues are highlighted in purple." Not purple but yellow.

Swuec *et al.* "Cryo-EM structure of cardiac amyloid fibrils from an immunoglobulin light chain (AL) amyloidosis patient",

Detailed answers to Reviewers

Reviewer #1 (Remarks to the Author):

a) The work by Swuec *et al* describes the 4.0 Å resolution structure of an amyloid fiber by CryoEM. This report is very relevant to the field of conformational diseases because the fiber was obtained *ex vivo* from the heart of a patient with amyloidosis. The description of the structure is very clear and shows interesting features, in particular, the transformation of the native b barrel of the light chain into planar fibrils also formed by b strands. My only concern with the article is the lack of reference and comparison with previous *in vitro* work about the folding and fibril formation of the protein encoded by the same germ line.

a) >> *Although not directly requested, a sentence at page 5, l.2 (line 97 of the original manuscript) has been amended to a more specific statement on the germline gene: "AL55 is of λ isotype and belongs to the IGLV6-57 germline gene, which is overrepresented in the repertoire of amyloidogenic LCs, compared to the polyclonal repertoire^{24,25}."*

b) The study of protein unfolding *in vivo* is very challenging, nevertheless there is abundant experimental information about the unfolding and refolding of this particular germline protein *in vitro*, for example, the authors state (p.11 end of first paragraph) that "energetic considerations suggest that such unfolding should not occur when the protein is in a monomeric state" in this context, the energetics (and kinetics) of the unfolding of the germline protein (Blancas-Mejia *et al* J.Mol.Biol. 2009), are relevant.

b) >> *The original sentence (line 219) "energetic considerations suggest that such unfolding should not occur when the protein is in a monomeric state" was meant to imply that in principle the unfolding process is likely to be energetically more favored on the surface of growing fibrils than in solution surrounded by water molecules. Since no data in our work indeed provide experimental evidence on this aspect, and moreover all three Reviewers consider this sentence either unclear or speculative, it has been removed.*

ACTION: SPECULATIVE SENTENCE REMOVED.

c) Likewise, the authors describe the role of residues and regions of the protein, such as Pro7 and the N-terminal strand, whose role in stability and fibrillogenesis have been addressed *in vitro* (Hernández Santoyo *et al* J. Mol. Biol. 2010), (del Pozo Yauner *et al* Biochem.Biophys.Research.Comm. 2014). Such information is also available for proteins derived from patients (González-Andrade *et al* FEBS J 2013).

One of the more relevant points of the work by Swuec *et al* is that it shows a new structure with the adequate resolution to bring a step forward the transformation between the globular native fold and the amyloid fibril. In this context, a comparison of *in vivo*, *ex vivo*, *in vitro* and *in silico* approaches should be very fruitful. As stated by the authors, the differences between fibrils obtained from patients and those obtained *in vitro* or in model systems is an open question, therefore, a comparison with the NMR data recently reported for the fibrils formed *in vitro* for the same allotypic variant (Lecoq *et al* ChemBioChem 10.1002/cbic.201800732) should be relevant.

c) >> *Reviewer 1 has correctly pointed at a number of thorough biophysical papers, which are relevant for the discussion on the nature/structure of natural vs. in vitro grown amyloid fibrils. All the suggested biophysical papers have been now discussed at pg 13-15, while the paper by Lecoq *et al.* (published after our manuscript had been submitted) at pg15.*

ACTION: the following sentences have been added/amended.

- (Pg. 14 of amended ms.) "LC sequence and fibril assembly

AL55 sequence belongs to the $\lambda 6$ subgroup and in particular to the IGLV6-57 germ line gene, which is expressed in about 2% of in bone marrow plasma cells expressing λ light chain²⁵ but makes up to 18% of all the monoclonal λ proteins responsible for AL amyloidosis²⁴. Given the relevance of such subgroup in AL pathogenesis, several previous studies have analyzed proteins belonging to the IGLV6-57 segment, their aggregation propensity and the role of specific positions in tuning protein stability and amyloidogenicity²⁹. In particular, the interaction between residues Phe2 and Arg25 highly stabilizes the V₁ native fold, while amino acid variations in one of the two positions increases the overall aggregation propensity^{30,31}. Gly in position 25, as occurring in AL55, is reported to facilitate amyloid formation³¹.
(NEW REFERENCES 29-31 ADDED)

- (Pg 15) “Frequently, an extra prolyl residue (Pro8) can be found, which would be located in the $\beta 1$ - $\beta 2$ turn, not impairing the fibril fold observed for AL55. Indeed, previous evidence showed that the His8→Pro mutant in V₁ domain belonging to the IGLV6-57 slowed but did not abrogate fibril formation³².”
(NEW REFERENCE 32 ADDED)

- (Pg 15-16) “A recent ssNMR model of *in vitro* fibrils formed by a V₁ belonging to the IGLV6-57 gene segment and with only 12 mutations when compared to AL55 sequence, shows several structural analogies. Even though the N-terminal stretch is predicted partially flexible, both the reported polymorphs display two ordered regions (residues 20-45 and 65-103) with parallel arrangement and a disordered region spanning residues 45-60³³.”
(NEW REFERENCE 33 ADDED)

Reviewer #2 (Remarks to the Author):

The paper by Swuec et al presents a 4.0 Å resolution cryo-EM structure of AL55 amyloid fibrils composed by two distinct polypeptide stretches in a total of 77 residues, which were extracted from a patient’s heart. This work is novel as, in spite of the fact that there are currently a number of published structures of amyloid fibrils (amyloid-beta, tau, synuclein,...), this is one of the first reported structures for an AL amyloid fibril, and the field still needs to build up knowledge on this type of protein structures to better understand and mitigate its formation. Therefore, the information within this paper has a broader interest to the protein science and structural biology communities. The work is technically proficient, and the reported structural analysis is competent. The aspects below require some consideration:

a) A scale bar is missing in Fig 1a

a) >> **ACTION** – Scale bar added and Fig.1 (panel a) legend amended: “Myocardial tissue from patient AL55, stained with Congo red. Red-orange stain and apple-green birefringence indicate amyloid deposits under visible (left) and under polarized light (right), respectively (magnification 100X; scale bar 100 μm).”

b) In vivo unfolding is suggested to be an obligate step in fibril formation. At some point the authors state that “...energetic consideration suggest that unfolding should not occur ...in the monomeric state” (page 11, Lines 219-200). However, this needs to be either revised or consubstantiated with data as no evidence is provided to support this claim.

b) >> *The original sentence (line 219) “energetic considerations suggest that such unfolding should not occur when the protein is in a monomeric state” was meant to imply that in principle the unfolding process is likely to be energetically more favored on the surface of growing fibrils than in solution surrounded by water molecules. Since no data in our work indeed provide experimental evidences on this aspect, and moreover all three Reviewers consider this sentence either unclear or speculative, it has been removed.*
ACTION: SPECULATIVE SENTENCE REMOVED.

c) Subtitle ‘AL55 unfolds along the aggregation pathway’ is misleading in face of presented results in this paper, as the aggregation pathway is not studied. Paraphrasing the authors, this section should be changed to ‘AL55 unfolding is an obligate step for amyloid formation’- p10, line 202

c) >> *ACTION: the subtitle was amended to “AL55 unfolding is an obligate step for amyloid formation”*

d) Proteolysis of AL55 as cause for amyloid formation stems from an inference from the author’s data and is therefore essentially a working hypothesis/speculation. Therefore, this last section from results should be toned-down and a shorter version of this possibility can be rather (briefly) presented within the conclusions.

d) >> *As Reviewer 2 points out, the paragraph on the role of proteolysis is rather speculative. Nevertheless, we believe it holds value in this context, since our results allow drawing some concepts on the role of proteolysis. As suggested by Reviewer 2, in the amended version this is no longer an independent section, but has been written as the last shortened paragraph of the “LC sequence and the fibril assembly” section; moreover, we more clearly stated its meaning as a working hypothesis.*

ACTION – The Section has been amended as follows: (pg. 16-17 of amended ms.)

“Finally, some considerations on the role of proteolysis in AL amyloidosis may be drawn. Although the presence of multiple N-terminal LC fragments is a universal finding in AL amyloid fibrils, it is however unclear whether proteolysis releases amyloidogenic LC fragments, which then assemble into fibril deposits, or whether proteolysis occurs after amyloid formation. Recent reports suggest that susceptibility to proteolysis is distinctive for amyloidogenic LCs^{6,34,35}. Our nLC-MS/MS analysis of AL55 fibrils allowed the identification of peptides from LC fragments extending to the distal ends of the first or of the third C₁ domain strands. These protein regions are solvent exposed in the native LC domain structure (Supplementary Information Fig. 1), thus proteolysis could feasibly take place when the LC chains are natively folded. In turn, such cleavages may well destabilize (*i.e.* start unfolding of) the C₁ domain, whose structural integrity is known to play a stabilizing role for the full LCs³⁶. The overall structure of the AL55 fibrils on the other hand, shows that the C₁ domain is not protected in the mature fibrils, thus it might be completely removed by proteolysis occurring on fibril deposits. Taken together, the above structural and biophysical considerations allow speculating that LC proteolysis may occur to a large extent before aggregation.”

Reviewer #3 (Remarks to the Author):

The manuscript by Swuec et al. describes a cryo-EM structure of a fibril formed by immunoglobulin light chains (LC), which were extracted from the heart of a systemic light chain (AL) amyloidosis patient. This work describes so far only the second structure of an amyloid fibril sample that has been extracted from a patient (following the work on the tau fibril by Fitzpatrick et al). This is already a great achievement and makes the structure highly interesting. Furthermore, the atomic structure of the AL fibril is an important step for understanding the molecular foundation of AL amyloidosis. The structure determination (experiment and data analysis) seems sound and technically well done. The cryo-EM structure has a good resolution of 4.0 Å and the density looks sufficiently well defined to allow for building an atomic model with reasonable confidence. Interestingly, the fibril core contains a region of high sequence variability and the authors analyze convincingly how difference sequences could be accommodated by the presented fibril structure. A point that is very interesting for the amyloid field in general is the fact that the LC is natively folded and the study shows that the protein has to completely refold to fold into the fibril structure, since none of the side-chain interactions are conserved and they differ in the fibril completely from those

in the native structure. The manuscript is very well written and results are clearly presented. I think the manuscript is very well suited for publication in Nature Communications.

Minor points:

a) p.11: "while about one third of the fibril VI (residues 66-105) fall in a poorly structured region." The assignment "residue 66-105" seems to be a typo. As shown further below, they probably refer to the residues 38-65 und around 105.

a) >> *ACTION – the typo was amended accordingly to: “(residues 38-65)”.*

b) p.11 "moreover, energetic considerations suggest that such unfolding should not occur when the protein is in a monomeric state but later on, along the aggregation pathway." A reference or more explanation should be added, it is not clear to me what this means.

b) >> *The original sentence (line 219) “energetic considerations suggest that such unfolding should not occur when the protein is in a monomeric state” was meant to imply that in principle the unfolding process is likely to be energetically more favored on the surface of growing fibrils than in solution surrounded by water molecules. Since no data in our work indeed provide experimental evidences on this aspect, and moreover all three Reviewers consider this sentence either unclear or speculative, it has been removed.*

ACTION: SPECULATIVE SENTENCE REMOVED.

c) Figure 5c "Conserved residues are highlighted in purple." Not purple but yellow.

c) >> *ACTION – color coding has been amended accordingly*